# Primary Teeth-Derived Demineralized Dentin Matrix Autograft for Unilateral Maxillary Alveolar Cleft during Mixed Dentition

**DOI:** 10.3390/jfb13030153

**Published:** 2022-09-15

**Authors:** Yusuke Matsuzawa, Naoto Okubo, Soichi Tanaka, Haruhiko Kashiwazaki, Yoshimasa Kitagawa, Yoichi Ohiro, Tadashi Mikoya, Toshiyuki Akazawa, Masaru Murata

**Affiliations:** 1Department of Oral and Maxillofacial Surgery, Graduate School of Dental Medicine, Hokkaido University, Sapporo 060-8586, Japan; 2Division of Oral and Maxillofacial Surgery, Keiyukai Sapporo Hospital, Sapporo 003-0026, Japan; 3Laboratory of Molecular and Cellular Medicine, Faculty of Pharmaceutical Sciences, Hokkaido University, Sapporo 060-8586, Japan; 4Division of Maxillofacial Diagnostic and Surgical Sciences, Faculty of Dental Science, Kyushu University, Fukuoka 812-8582, Japan; 5Department of Oral Diagnosis and Medicine, Graduate School of Dental Medicine, Hokkaido University, Sapporo 060-8586, Japan; 6Industrial Technology and Environment Research Development, Hokkaido Research Organization, Sapporo 060-0819, Japan; 7Division of Oral Regenerative Medicine, School of Dentistry, Health Sciences University of Hokkaido, Tobetsu 061-0293, Japan

**Keywords:** alveolar cleft, bone graft, chin bone, DDM, primary tooth, teeth, dentin

## Abstract

This clinical report describes the immediate autograft of primary (milk) teeth-derived demineralized dentin matrix (DDM) granules for a 6-year-old boy with unilateral alveolar cleft. First, four primary teeth were extracted, crushed in an electric mill for 1 min, and the crushed granules were demineralized in 2% HNO_3_ solution for 20 min. Simultaneously, the nasal mucoperiosteum was pushed upwards above the apices of the permanent central incisor adjacent to the cleft. The nasal and palatal openings were closed by suturing the mucoperiosteum on both sides of the cleft with absorbable threads. The wet DDM granules were grafted into the managed cleft triangle space, and a labial flap was repositioned. The radiographic images at 6 months showed the continuous hard tissues in the cleft area and DDM granules onto lateral incisor (22) and impacted canine (23). The 3D-CT views at 2 years showed impacted tooth (22) blocked by primary canine and the replacement of DDM granules by bone near teeth (22,23). At 4 years, tooth crown (22) was situated just under the mucous membrane, and teeth (22,23) erupted spontaneously until 6 years without a maxillary expansion and a tow guidance of canine. The DDM granules contributed to bone formation without the inhibition of spontaneous tooth eruption. We concluded that autogenous primary teeth DDM graft should become a minimally invasive procedure without bone harvesting and morbidities for unilateral alveolar cleft.

## 1. Introduction

Secondary autogenous cancellous bone graft is a widely used technique for the treatment of alveolar clefts, and it has been performed preferably before the eruption of the permanent canine in order to provide periodontal support for the eruption and preservation the teeth adjacent to the cleft [1,2]. The donor site of the harvesting autogenous bone is primarily the iliac crest, that provides an abundant volume of cancellous bone tissue. Since 1998 in Hokkaido University Hospital, a mono-cortical mandibular bone graft was performed, and the chin bone and/or mandibular body have been donor sites for alveolar cleft bone grafting in the mixed dentition stage [3]. During harvesting chin bone, surgeons must consider the incisive nerve, the unerupted mandibular canine, and the integrity of the inferior mandibular border. Although the harvesting bone from the chin is easy and safe, the volume of the harvested chin bone is little in the presence of an unerupted mandibular canine; thus, an autogenous chin bone was recommended for late alveolar bone grafting [4]. Donor site morbidity causing the need to harvest autogenous bone is an important factor for a patient’s quality of life. To avoid the donor site morbidity, we focused on an autogenous primary teeth-derived dentin matrix. The donor site morbidity, pain, and prolonged hospitalization have prompted the search for bone graft materials. Since 2015, primary teeth have been recycled as demineralized dentin matrix (DDM) granules for unilateral alveolar cleft regeneration in Hokkaido University Hospital as clinical studies (approval code No.011-0030).

Dentin and bone are biologically similar, consisting of apatite (70%), collagen (18%), non-collagenous proteins (NCPs: 2%), and body fluid (10%) in weight volume [5]. A bone-inducing property of rabbit demineralized bone matrix (DBM) was discovered in 1965 [6], and the same activity of rabbit DDM was also reported in 1967 [7,8]. Fresh and properly prepared DBM and DDM performed better in bone induction than highly calcified tissues such as normal calcified dentin and cortical bone [8,9,10]. It was considered that apatite crystals inhibited the release of bone morphogenetic proteins (BMPs) along with growth factors in the mineralized matrix. From the beginning of the 21st century, permanent tooth-derived materials have been applied as several forms of DDM for bone regeneration [11,12,13,14,15,16,17,18,19,20]. Until now, almost all cases were bone augmentation for implant placements [12,13,14,15,16,17,18,19], and a few were guided bone regeneration (GBR) for tooth transplantation [11,20]. Based on the evidences of the similarity in the components of dentin and bone, DDM have been applied widely for bone regeneration such as DBM.

In the present case, the source of DDM was a patient’s primary teeth. Four primary teeth in a 6-year-old boy were recycled as the granular type of DDM materials, and the immediate autograft of DDM granules was carried out for the bone bridge formation in the unilateral alveolar cleft and the eruption of upper lateral incisor (22) and canine (23). The aim of the DDM graft was to estimate bone formation related with the eruption of permanent teeth (22,23) clinically and radiologically for the follow-up of 6 years.

## 2. Materials and Methods

### 2.1. Patient

A 6-year and 3-month-old boy presented with non-syndromic unilateral cleft alveolus in the mixed dentition stage (Figure 1 and Figure 2A,B). As medical history, he was born in 2010 with normal delivery in an obstetrics and gynecology clinic (gestational age: 41 weeks, body weight: 3600 g), and he presented with non-syndromic unilateral cleft lip and alveolus. At 1 month old, he visited the department of oral and maxillofacial surgery in our hospital of Hokkaido University with his parents (mother: 27 years old, father: 31 years old), and at 4 months old, cheiloplasty (lip closure) was performed in the department of plastic surgery. Initial X-ray photos including computed tomography (CT) were taken before the DDM graft surgery in 2016. The width of the cleft was measured by the CT as 5.41 mm (alveolar top) and 8.23 mm (alveolar basal).

### 2.2. Primary Teeth Extraction

Primary teeth (52, 51, 61, 62) were extracted under the general anesthesia in 2016, based on the oral examination including CT and the treatment planning. A root absorption (length: 3–4 mm) in tooth 51 was recognized (Figure 2C).

### 2.3. Preparation of Demineralized Dentin Matrix (DDM)

Four primary teeth were crushed with saline ice blocks in a Zirconium (ZrO_2_) vessel with a ZrO_2_ blade at 12,000 rpm for 1 min by an electric mill (OSTEO-MILL^TM^, WiSM Mutoh Co., Ltd., Tokyo, Japan) (Figure 2D). The crushed teeth granules were immediately demineralized in 1.0 L of 2.0% HNO_3_ solution for 20 min [20]. The DDM granules including cementum were extensively rinsed in cold distilled water for 10 min (Figure 2E).

### 2.4. Surgical Procedure of Unilateral Alveolar Cleft and Autograft of DDM

Wide exposure of the cleft area was achieved through incisions. The nasal mucoperiosteum was pushed upwards above the apices of the permanent central incisor adjacent to the cleft. The nasal and palatal openings were closed by suturing the mucoperiosteum on both sides of the cleft with absorbable threads (5-0 Polysorb^®^; Covidien, Mansfield, MA, USA) (Figure 3A). The wet DDM granules were grafted into the cleft triangle space (Figure 3B), and the labial flap was repositioned and sutured with the same absorbable threads (Figure 3C,D). The surgical time including the preparation of DDM was 74 min. The wound area was covered with a sterilized gauze including antibacterial ointment (bacitracin: BC), and it was protected by a custom-made acrylic plate for 1 week. The patient took antibiotics as follows: cefazolin sodium (CEZ: 500 mg, 2 times/day) for 2 days and then amoxicillin (AMPC: 6 g, 3 times/day) for 3 days.

### 2.5. Radiographic Evaluation

Panoramic X-ray photos (Beraviewepocs type CR^®^; MORITA, Tokyo, Japan) and dental X-ray photos (ALULA-TS^®^; ASAHIROENTOGEN, Kyoto, Japan) including medical CT (Aquilion Prime SP^®^; Canon, Ohtawara, Japan) were taken sequentially after the surgery, and then, the eruption of the lateral incisor and canine was evaluated by clinical and X-ray photos. The 3D images were acquired by a spatial resolution soft (OsiriX Lite^®^; Softonic, Barcelona, Spain).

### 2.6. Orthodontic Treatments

The orthodontic wire with brackets was set in the maxillary teeth, and the orthodontic treatment started on July 2017, and it continued until now. In 2022, the lingual arch was set for maxillary forward pull. Both a maxillary expansion and a tow guidance of canine were not applied.

## 3. Results

### Gross View and Radiographic Evaluation after Surgery

One day after the surgery, a blood clot was formed in the extracted sockets. Mucosa inflammation at the recipient site was slight, and wound healing proceeded normally. The maxillary splint was removed at 7 days. The grafted DDM granules were not exposed. X-ray CT photos were taken at 6 months after the DDM graft (Figure 4), 2 years (Figure 5) and 4 years (Figure 6). During the whole follow-up period for 6 years, complications did not occur. In the CT images at 6 months, a continuous contour of maxilla was observed (Figure 4A,B). The grafted DDM were confirmed near teeth (22,23) as radiopaque granules and could be distinguished from the original bone by the structure and the radio-opacity (Figure 4C,D).

The 3D-CT views at 2 years after DDM graft showed a tendency of the eruption of impacted tooth 22 (Figure 5A,B). Tooth 22 before eruption was inclined distally, and tooth 22 was contacted with primary tooth 63 (Figure 5A). DDM granules were not found near teeth (22,23) (Figure 5C,D), and the smooth cortical line of alveolar bone was observed clearly (Figure 5C–E). In addition, normal spaces around the crown of tooth 23 were seen (Figure 5D,E).

At 4 years after surgery, tooth 22 was under mucosa and just before eruption (Figure 6A–C), and canine (23) was still impacted (Figure 6D). At 6 years, teeth (22,23) erupted spontaneously, the canine (23) was contacted to lower premolar (Figure 7A,B). Maxilla view of mirror image showed a smooth arch (Figure 7C), and the periodontal ligament space of tooth 22 was seen (Figure 7D).

## 4. Discussion

Alveolar cleft repair has conventionally relied on autogenous fresh iliac cancellous bone graft. In the follow-up, 36 of the 50 patients (72%) had canines spontaneously erupt through the grafted iliac bone granules, indicating a strong tendency for spontaneous tooth eruption after the secondary bone graft [2]. Especially, the complication rate following the harvesting of iliac bone was reported to be 19.3% [21].

### 4.1. Primary Teeth-Derived DDM

Since a bone-inducing property of rabbit DDM granules was reported in 1967 [7,8], a first clinical trial of permanent teeth-derived DDM autograft was achieved for bone augmentation of sinus floor in 2002 [22]. In this century, the main sources of DDM are derived from wisdom teeth. The permanent teeth-derived DDM have been applied for bone regeneration. In the present case, the sources of DDM granules were autogenous primary teeth. Thus, a donor site for bone harvesting was not required. Lateral incisor and canine nearby the cleft spontaneously erupted at 4–6 years post-operatively through the grafted DDM area (Figure 6 and Figure 7), indicating that DDM were harmonized with host and replaced by bone without inhibition of spontaneous tooth eruption. We believe, therefore, that the DDM granules created 3D-osteogenic spaces in the defect, and vascularization and osteoblast differentiation occurred between the biological granules. Bone grafting was principally performed to unify the maxilla, create an osseous environment that would support adjacent cleft teeth eruption into the arch, and close oral–nasal communication [3]. Our newly developed DDM method should become a minimally invasive procedure without bone harvesting and morbidities such as paresthesia and walking pain for children.

### 4.2. Biological Similarity between Dentin/DDM and Bone/DBM

Dentin is biologically similar to bone in its components [5]. DDM and DBM consist of acid-insoluble type I collagen and matrix-binding proteins such as BMPs and FGFs. DDM has a better performance in bone regeneration than calcified dentin [23]. The main advantages of DDM materials are as follows: (1) non-sacrifice of healthy donor site; (2) growth factors for bone regeneration; (3) cross-linked collagen; (4) non-immune rejection; (5) absorbability; (6) Arg-Gly-Asp (RGD) sequence; and (7) hydrophilic. On the other hand, the limitation of DDM volume was picked up as a disadvantage. In our case, the volume of wet DDM derived from four primary teeth was adequate for alveolar cleft defect (width of alveolar top: 5.41 mm, alveolar basal: 8.23 mm). For large alveolar bone defects, autogenous DDM alone surgery cannot be applied. To address this issue of a larger cleft defect, the parent’s wisdom teeth-derived DDM graft will be considered for their cleft child, because acellular DDM graft is a matrix-based therapy unlike a cell therapy. Already, an allograft of DDM between family were performed first in Korea. The DDM granules were derived from a child’s wisdom teeth for a parent’s implant placements [24].

### 4.3. Morbidities Related with Bone Harvesting

The mandibular bone (chin or body) graft in early mixed dentition has the possibility of resulting in necrosis of the pulp or devitalization of teeth, and injury to the mental nerve could result in objective and subjective disturbances in the sensitivity of adjacent teeth and soft tissues [25,26]. The timing of bone harvesting usually overlaps with a patient’s school days and limits the patient’s sporting activities. Recently, morbidities (paresthesia and walking), duration of surgery, and the bone formation in alveolar defects were compared between the chin bone graft and iliac crest bone graft [27]. The results revealed that normal walking (1 day) and surgery time (40 min) in the chin bone group were significantly faster than those (9.5 days, 76 min) in the iliac bone group, while bone formation in alveolar defects was almost the same at 12 months after the surgery in both groups.

### 4.4. New Methods for Bone Formation in Alveolar Cleft

Interestingly, new methods were reported for bone bridge formation in alveolar cleft [28,29,30,31,32]. In 2006, bone marrow-derived stem cells (BMDSCs) were isolated from a patient’s healthy iliac crest by bone marrow aspiration (20 mL) and cultured for about one month for the preparation of tissue-engineered osteogenic material, which is named as injectable bone [28]. After the soft tissue management of the cleft, the defect was supported with a titanium mesh plate (thickness:1.0 mm), and then, the prepared tissue-engineered osteogenic material was injected into the pouch through a syringe. The cell-based therapy is indeed a tissue-engineered method, but a high cost and long time period for cell culture in cell processing center was pointed out, and the titanium mesh plate must be removed for tooth eruption.

In 2010, lateral cortical bone plates from the chin and/or body were grafted on the labial and palatal openings of the alveolar process defect without particulate bone [3]. At 6 months postoperatively, CT analyses showed sufficient bone bridge formation at the cleft site in 85.4% of clefts, and the cleft-adjacent canines erupted spontaneously in 92.6% of clefts. The mono-cortical mandibular bone grafting could reduce the volume of harvesting bone and should be applied for large clefts as advantages.

In 2013, a maxillary bone transport technique was combined with three supernumerary teeth-derived DDM graft [29]. During the removal of the bone transporter, the graft of DDM granules to the docking site might be an effective approach for large alveolar cleft repairs under local anesthesia without any hospital stays.

In 2017, cone beam computed tomography (CBCT) analyses demonstrated comparable bone regrowth and density values following secondary alveolar cleft repair using bone morphogenetic protein-2 (BMP-2)/DBM scaffold versus autologous iliac bone graft [30]. Additionally, BMP-2/DBM material was grafted to reconstruct defects related to clefts in three patients [31]. After 3–7 months, all patients had generated bone in the clefts and did not require bone grafting. The articles highlighted that combining BMP-2 and DBM in higher risk patients is an excellent option to avoid bone graft loss and reoperation [30,31].

In 2022, octacalcium phosphate (OCP)/atelocollagen composite disk is available for bone defects and alveolar cleft repair in Japan [32]. OCP is known to be a precursor of biological apatite and promotes bone formation [33]. Although the OCP/collagen disk does not include growth factors, the bone bridge in all four patients in the OCP/collagen group was successfully formed in the alveolar cleft, and by 6 months postoperatively, the permanent teeth had erupted at the OCP/collagen graft area. The OCP/collagen disk is a highly porous and biomimetic material that is similar to the components of dentin and bone.

### 4.5. Near Future of DDM

Vital tooth-derived dentin is an extracellular matrix with growth factors and without cells. The application of a cellular DDM is a matrix-based therapy, unlike a cell-based therapy. Therefore, immunosuppressive drugs are not needed at all for the patients even after the allograft of DDM, although patients after liver transplantation must take immunosuppressive drugs. For children with alveolar clefts, we suggest that parents had better keep their wisdom teeth in the mouth and/or stock their DDM in the teeth bank under the guidance of dental doctors. Being larger in quantity, parent-derived DDM has the potential to compensate for the small amount of autogenous primary teeth DDM. Moreover, DDM combined with BMP-2 will facilitate bone formation in higher risk patients [34].

## 5. Conclusions

Autogenous primary teeth DDM granules were harmonized with the host tissues and replaced by bone, without both interference of tooth formation and inhibition of spontaneous tooth eruption. In this case, we have showed that autogenous primary teeth DDM graft is a viable and minimally invasive procedure without morbidities and might be a valuable alternative to autogenous bone graft for unilateral alveolae cleft.

## Figures and Tables

**Figure 1 jfb-13-00153-f001:**
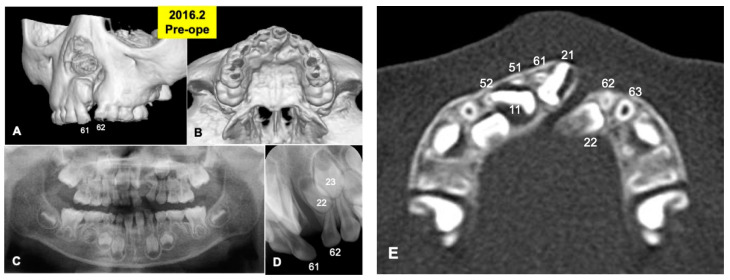
X-ray photos in 2016 before surgery. (**A**) Labial view of 3D-CT showing unilateral alveolar cleft. (**B**) Palatal view of 3D-CT. (**C**) Panoramic view in mixed dentition stage of 6-year-old boy. Congenital missing teeth: 12,21,25. (**D**) Occlusal view indicating unilateral cleft between primary teeth 61 and 62 (permanent teeth 21 and 22). (**E**) Axial CT photo. Coronal level of cleft-adjacent incisor (21). Images of primary teeth-roots (52, 51, 61, 62) before extraction.

**Figure 2 jfb-13-00153-f002:**
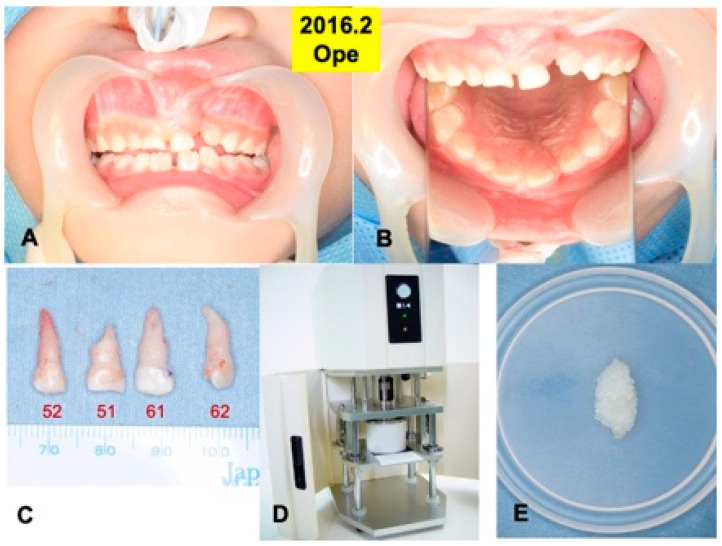
Surgical procedure in 2016. (**A**) Appearance just before surgery. (**B**) Mirror image. (**C**) Extracted primary teeth (52, 51, 61, 62). (**D**) Electric mill for tooth crush (OSTEO-MILL^TM^, WiSM Mutoh Co., Ltd., Tokyo, Japan) (**E**) Wet DDM granules before use.

**Figure 3 jfb-13-00153-f003:**
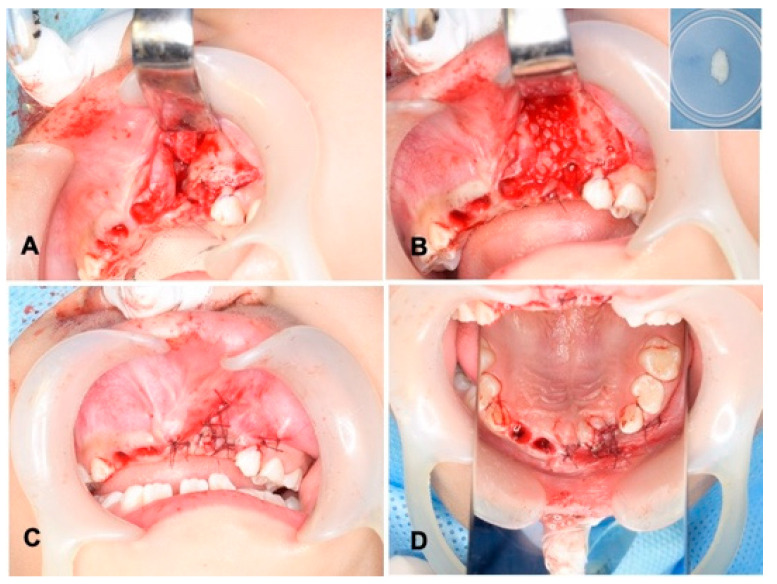
Surgical procedure. (**A**) Exposed alveolar triangle defect and extracted sockets. (**B**) DDM graft after graft-bed management (closure of nasal lining with mucoperiosteum). Right upper: DDM granules. Note: DDM and blood coagulation. (**C**) Flap repositioned and sutured with absorbable threads. (**D**) View of mirror image.

**Figure 4 jfb-13-00153-f004:**
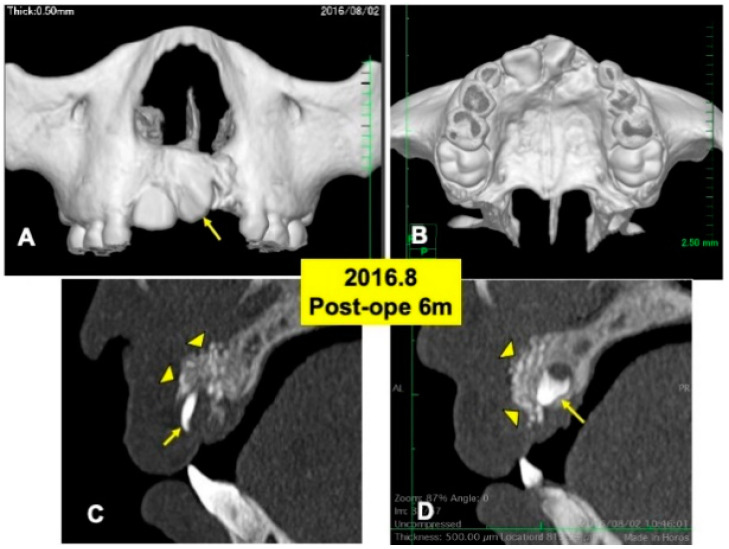
X-ray CT photos at 6 months after surgery. (**A**) Labial view of 3D-CT showing continuous hard tissues in cleft area. Arrow indicating rotation of tooth 21. (**B**) Palatal view of 3D-CT. (**C**) DDM granules (arrowheads) in cleft near incisor (22: arrow). (**D**) DDM granules (arrowheads) on impacted canine (23: arrow).

**Figure 5 jfb-13-00153-f005:**
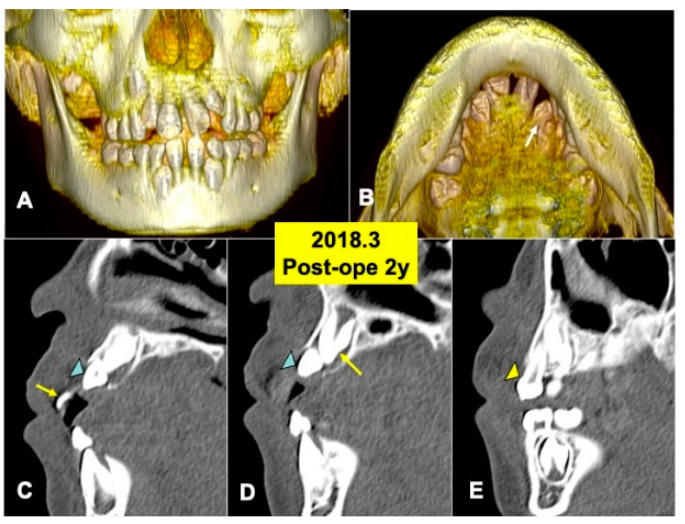
X-ray CT photos at 2 years after surgery. (**A**) Three-dimensional (3D)-CT image. Eruption of teeth 11 and 21. (**B**) Three-dimensional (3D)-CT image. Arrow indicating tooth 22. (**C**) Sagittal appearance near teeth (22,23). Arrow indicating tooth 21. Arrowhead indicating tooth 22. (**D**) Sagittal appearance near teeth (22,23). Arrow indicating tooth 23. Arrowhead indicating tooth 22. Note: Flat bone contour and no appearance of DDM granules. Arrowhead indicating tooth 23. (**E**) Sagittal appearance near teeth (63, 23). Arrowhead indicating primary canine (63).

**Figure 6 jfb-13-00153-f006:**
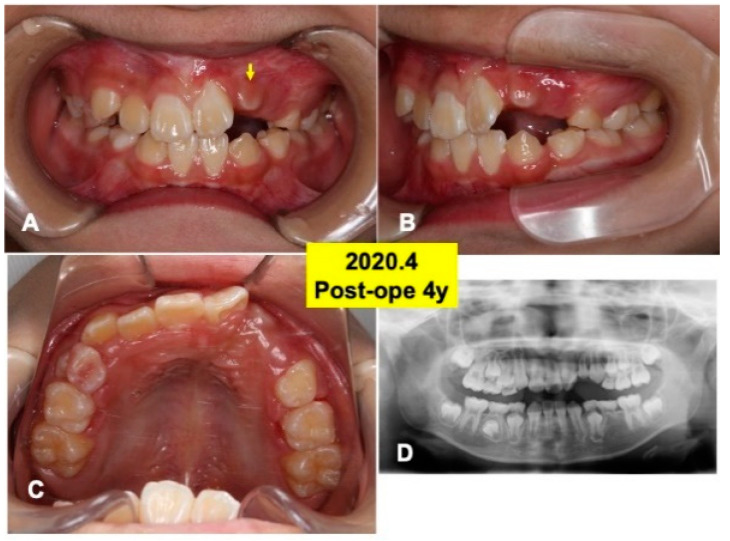
Intraoral views and panoramic X-ray photo at 4 years after surgery. (**A**) Appearance in 2020. Arrow indicating tooth 22 just before eruption. (**B**) Lateral appearance. Note: spontaneous shedding of primary canine (63) and impacted canine (23). (**C**) View of mirror image. (**D**) Appearance of X-ray photo. Adequate direction of tooth axis (22,23). Note: eruption of lower canine (33,43).

**Figure 7 jfb-13-00153-f007:**
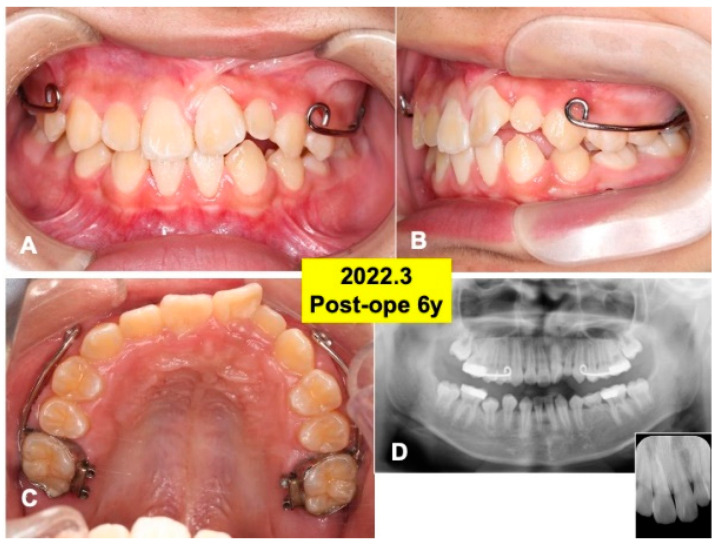
Gross views and panoramic X-ray photos at 6 years after surgery. (**A**) Appearance in 2022. Note: eruption of teeth (22,23). (**B**) Lateral appearance during orthodontic treatment. (**C**) View of mirror image. Set of maxillary protraction appliances. (**D**) Appearance of panoramic and dental X-ray photos. Note: healthy alveolar bone around teeth (22,23).

## Data Availability

Data sharing is not applicable to this article.

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
