# Peer review of "Primary Teeth-Derived Demineralized Dentin Matrix Autograft for Unilateral Maxillary Alveolar Cleft during Mixed Dentition"

_jfb, 2022, doi:10.3390/jfb13030153_

Round 1
Reviewer 1 Report
The recommended amendments are listed in the attached PDF file- "Amendments to JFB manuscript (01

Author Response
We would like to thank you very much for your valuable advice.
Our paper was revised under your guidance, except for Line 97.
We have used “root absorption” until now. “Re” means again.
So, we have selected “absorption”, not resorption in our papers.
Reviewer 2 Report
This article is interesting and informative for pediatric dentistry community, it is very well written and provides useful information that deserve publication.
I can't find any major flaws, but some minor changes should be done to rise the quality of the article if they are taken in the consideration.
Suggestion - replace the term milk with primary throughout the whole text
line:
97 Suggestion - replace the term absorption with resorption
113 which ointment – (name,manufacturer...)
114 CEZ - full name for the first appearance, then abbreviation
115 AMPC - full name for the first appearance, then abbreviation
Author Response
Our paper was brushed up and revised under your guidance.
Milk was replaced by primary.
We have used “root absorption” until now. “Re” means again.
So, we have selected “absorption”, not resorption in our papers.
Reviewer 3 Report
Journal: Journal of Functional Biomaterials (ISSN 2079-4983)
Manuscript ID: jfb-1915408
Type: Case Report
Title: Milk teeth-derived Demineralized Dentin Matrix Autograft for Unilateral Maxillary Alveolar Cleft During Mixed Dentition
This manuscript was intended to present the long term observations of a case of unilateral alveolar cleft treated with alveolar osteoplasty performed with milk teeth-derived demineralized dentin matrix autograf.
I was glad to read this concise and well written article which in my opinion fully comply with the requirements of that kind of report.
Only a few remarks:
In the introduction: ‘As the donor sites of the harvesting autogenous bone, iliac or chin bone have been regarded as a standard.’
- The fact that iliac cancellous bone graft remains the gold standard for alveolar bone grafting has been mentioned several times in the articles in your reference list.
What’s more the next sentence in the Introduction: ‘Especially in the mixed dentition stage, the chin bone is a common donor site for alveolar cleft bone grafting.’ Can be challenged by your reference number 4 – ‘Considering the fact that the volume of the harvested bone is low, especially in the presence of unerupted mandibular canine, autogenous chin bone is recommended for late ABG’. The sentence in the Introduction has to be corrected.
The problem comes back in the Discussion section: ‘Iliac and chin bone are popular donor site for bone harvesting for alveolar cleft repair.’ – may be in your center or why not to mention the tibial bone, cranial bone etc.?
The same sentence cannot be repeated twice in the article.
Author Response
We would like to thank you very much for your valuable advice.
<Main points>
1.Our paper was brushed up in Introduction especially about [3],[4].
2.Line 186. Repeated sentence was deleted.
- The use of tibial bone and cranial bone include ethically issues from the point of minimally invasive surgery. Therefore, we don’t mention the harvest of tibial bone and cranial bone for unilateral alveolar cleft patients.
Would you please check red sentences in our revised version ?
Reviewer 4 Report
To author:
Review for Manuscript ID: jfb-1915408
entitled " Milk teeth-derived Demineralized Dentin Matrix Autograft for Unilateral Maxillary Alveolar Cleft During Mixed Dentition”
The manuscript is of interest and has merit for publication. Just a few points that need to be corrected as follows:
1- Figure 2 blurred; better quality recommended.
2- Line 102, the bold at the beginning of the sentence “The crushed teeth” need to be removed.
3- Line 114 and 115; names of antibiotics should be written in full.
4- Line 202: define RGD.
Regards,
Author Response
We would like to thank you very much for your valuable advice.
Our paper was brushed up and revised under your guidance.
Would you please check red sentences in our revised version ?